**Data Availability Statement:** Data cannot be publicly shared because there are ethical.

# Policymaking through a knowledge lens: Using the *embodied-enacted-inscribed* knowledge framework to illuminate the transfer of knowledge in a mental health policy consultation process – A South African case study

**Debra Leigh Marais** [1]◉*, **Inge Petersen** [2]‡, **Michael Quayle** [3,4]‡

1 Research Development and Support Division, Faculty of Medicine and Health Sciences, Stellenbosch University, Cape Town, South Africa, 2 Centre for Rural Health, College of Health Sciences, University of KwaZulu-Natal, Durban, South Africa, 3 Centre for Social Issues Research, Department of Psychology, University of Limerick, Limerick, Ireland, 4 Department of Psychology, School of Applied Human Sciences, University of KwaZulu-Natal, Pietermaritzburg, South Africa

◉ These authors contributed equally to this work.
‡ These authors also contributed equally to this work.
* debbiem@sun.ac.za

## Abstract

### Background

Marrying principles of evidence-based policymaking, with its focus on *what works*, with principles of consultative policymaking, with its focus on *what works for whom*, means finding ways to integrate multiple knowledge inputs into policy decisions. Viewed through the lens of the *embodied-enacted-inscribed* knowledge framework, policy consultation is a site of knowledge enactment, where the embodied knowledge enacted by individuals engages with the inscribed knowledge contained in policy documents, creating new forms of embodied and inscribed knowledge that move beyond these spaces.

### Aim

Using this knowledge framework, this study aimed to trace the movement of knowledge inputs through South Africa's mental health policy consultation summit.

### Methods

Breakaway group session transcripts from the national consultation summit were thematically analysed to identify the types of knowledge that participants explicitly drew on (experiential or evidence-based) during discussions and how these knowledge inputs were used, responded to, and captured.

### Findings

Findings suggest that there was little explicit reference to either evidence-based or experiential knowledge in most of the talk. While slightly more evidence-based than experiential knowledge

Restrictions on sharing the complete data set. The data comprises transcripts of audio recordings of a 2-day policy consultation event. While these recordings are (publicly) available on request from the national Department of Health through the Access to Information Act, the recordings include the mention of the names of some key participants at the event. Participants may also be potentially identifiable from their contributions and the way they have spoken about their roles and work in mental health in South Africa. Sharing the full data set would violate the principle of confidentiality. For researchers who meet the criteria for access to confidential data, the Biomedical Research Ethics Committee at the University of KwaZulu-Natal (South Africa) can be contacted to request access to the data (contact: Ms Anusha Marimuthu, +27312604769, BREC@ukzn.ac.za).

**Funding:** The authors received no specific funding for this work.

**Competing interests:** The authors have declared that no competing interests exist.

claims were made, this did not render these claims any more likely to be responded to or engaged with in group discussions, or to be inscribed in group recommendations.

## Discussion

The importance of designing participatory processes that enable optimal use of knowledge inputs in these enacted spaces is discussed.

## Conclusion

Attending to the specific ways in which knowledge is transformed and moved through a policy consultation process has the potential to enhance the value that consultation offers policymakers.

## Introduction

### Knowledge in policy

Policymakers tasked with developing mental health policies must balance a number of competing demands, including the need to develop policies that are applicable on a national level, while simultaneously addressing the idiosyncratic and contextual particularities associated with mental ill health at individual and local levels. Marrying the principles of evidence-based policymaking, with its focus on *what works*, with the principles of consultative policymaking, with its focus on *what works for whom*, means finding ways to integrate multiple knowledge inputs to incorporate these into policy decisions. In this sense, policymaking represents something of a knowledge problem for policymakers.

In low- and middle-income countries (LMICs), it is particularly important that evidence of effective strategies gained through research is used to inform policy so that limited resources available can be put to good use [1]. Evidence-based policymaking in mental health poses particular complexities. One of the strongest arguments against evidence-based policymaking is based on the privileging of certain forms of knowledge over others, a carry-over from evidence-based medicine and the hierarchy of evidence. The use of randomised controlled trial methodologies that privilege group-level outcomes over idiographic design, for example, neglects the importance of individual experience and the meanings of health problems for those who live with them [2, 3]. In the mental health field in particular, the danger of a reliance on the kinds of knowledge generated at the top of the evidence hierarchy is that it risks reductively negating the personal and interpersonal significance and meaning of experience [4].

The nature of evidence as generalisable knowledge may limit its ability to respond to contextual idiosyncrasies typical of a diverse country like South Africa, where health system governance is decentralised and there is great variability across provinces in terms of population needs and capacity of the system to respond to these needs [5, 6]. In addition, policy, like evidence, needs to balance the tension between general applicability and local relevance [7]. This highlights the importance of attending to contextually based knowledge as one way of enhancing policy development and implementation. As such, some have called for a shift in policymaking from an emphasis on *what works* (evidence), to *what works for whom* (evidence-in-context), and argue that attending to different forms of knowledge in the policy consultation arena may provide a link between these two [8, 9]. Much of the evidence about the effectiveness of interventions has been generated in high-income countries, and the generalisability of

such information from one context to another has been called into question. Furthermore, "in the search for generalisable knowledge, evidence on mental disorders tends to downplay or ignore cultural variations" [10] (p. 53). There is thus a need for alternative ways of thinking about knowledge in evidence-based policymaking [11].

## Knowledge in policy consultation

Knowledge transfer in evidence-based policymaking has received much attention, but the specific ways in which knowledge moves through a policy consultation process are still poorly understood. This study is positioned at the interface between evidence-based and consultative policymaking, focusing on processes through which knowledge is transferred to inform policy, with an emphasis on participation and the integration of multiple forms of knowledge in mental health policy development. This underscores the importance of knowledge 'management' in policy consultation, which includes attending to how knowledge contributions might be elicited, captured, and transferred during such processes. The challenge facing policymakers is that policy consultation, by its nature, involves a wide range of diverse kinds and forms of knowledge contributions, which must be balanced against different–sometimes competing–policymaking objectives. In order to optimise the value that each of these knowledge inputs might hold in enhancing policy outputs, the legitimacy of different kinds of knowledge contributions must be recognised. Policy consultation processes must therefore be designed in such a way as to enable these kinds of inputs.

One purpose of consultation is to allow for participation and representation of a wide range of views and experiences, which, in turn, requires balancing scientific evidence with experiential evidence. Along with shifts towards person-centred mental health care have come calls to enable greater representation of mental health care users and their families in the development of mental health policies [12–14]. Inputs from these stakeholders and from the public more generally during policy consultation are unlikely to be consistently in the form of factual or evidence-based knowledge, but rather as practical or experiential embodied knowledge [15, 16]. This may be difficult to codify and therefore capture in documented–and transferable– forms [17–19]. Doing so will mean finding ways of structuring policy consultation processes to not only enable multiple forms of knowledge to be elicited and receive attention, but also to be moved through the consultation process to inform policy.

However, the problem identified here is that the movement of embodied knowledge through the policy consultation process to inscribed knowledge is likely to be limited, particularly in more conventional consultation formats such as summits or public hearings [19]. This may be, in part, because this form of knowledge may not be considered as legitimate as input framed as 'evidence', and therefore not receive attention. It may also be because this type of embodied knowledge does not easily lend itself to abstraction, generalisation, and inscription, which is a necessary part of a policy consultation exercise. This study thus set out to trace and classify the types of knowledge that are produced at a consultation event, and to assess whether and how embodied knowledge moved into and between enacted and inscribed knowledge forms.

## Embodied-enacted-inscribed knowledge

Freeman and Sturdy's conceptual schema of knowledge functions in policy provide a theoretical lens through which knowledge and the points at which it is transferred and transformed can be better understood [7]. This schema distinguishes between three phases of knowledge, each analogous to three phases of matter–*embodied* (liquid), *enacted* (gas), and *inscribed* (solid)–that can be transformed between phases through various kinds of (inter)action.

Embodied knowledge is a form of knowledge that is "deeply embedded in bodily experience" [7] (p. 9). It is typically understood as tacit, practical, and experiential. However,

Freeman and Sturdy also incorporate explicit, factual, or theoretical knowledge in their definition of embodied knowledge. This more explicit, fact-based type of knowledge has sometimes been defined as *embrained*, and can be seen as representing, among other things, the evidence-based knowledge that individuals embody and are easily able to articulate in verbal form. The premise is that all knowledge, whether verbal or non-verbal, is experienced by and in the physical being of each individual. Embodied knowledge thus includes both experiential and evidence-based (factual/theoretical) knowledges that individuals embody.

Enacted knowledge is knowledge-in-action, the observable enactment of the other two knowledge forms. Like gas is to solid and liquid, enacted knowledge represents the transformation of embodied and inscribed knowledge into a new, active form. Freeman and Sturdy recognise enacted knowledge as a distinct form of knowledge which may, in turn, result in the production of new embodied or inscribed knowledge [7]. Like gas, enacted knowledge is transient and only visible through (inter)action. It is a particularly useful concept in relation to this study, as it calls attention to the interactive meeting spaces in which participants' embodied knowledge can be enacted.

Inscribed knowledge is a codified form of knowledge that is captured and encoded in material artefacts–in this study, primarily as words in documents. Inscribed knowledge represents a 'translated' (encoded and inscribed) form of embodied and enacted knowledge that is stable, easily reproduced, and easily transferable. Inscription of knowledge typically involves some form of abstraction in the codification process. In a policy consultation process, the inscription of knowledge in documents is a critical component of transferring knowledge inputs to policy outputs.

This conceptualisation of knowledge was considered particularly relevant for the current study, not only because it highlights the *movement* of knowledge through policy, but also as it does not presume that one form of knowledge precedes or is superior to another, nor is one form associated with a particular kind of actor in the policy space [7]. As such, it collapses the evidence-based knowledge hierarchy, and allows for due consideration of multiple knowledge inputs which, as highlighted above, is especially important in mental health and developing country contexts, particularly given the effects on human experience of differing notions of selfhood across cultures [10].

## Context of study

The aim of this study was to classify the types of knowledge that were produced at South Africa's national mental health policy consultation summit and to assess whether and how embodied knowledge moved into and between enacted and inscribed knowledge forms–in particular, into policy outputs. This was done using case study of how these different forms of knowledge were–or were not–used in a consultative policymaking process at a micro (individual) level. South Africa's *Mental Health Policy Framework and Strategic Plan 2013–2020* [20] was adopted in 2013. The development of this policy included an extensive consultation process, involving consultation summits in eight of the nine provinces and culminating in a national consultation summit, in April 2012, where input was invited on the draft policy document. The consultation process and follow through from provincial to national summits has been described in detail elsewhere [21]. The ten group breakaway sessions held over two days at the national consultation summit is the focus of the current paper.

The themes of the breakaway sessions are presented in Table 1, together with the abbreviations of these topics that are employed in subsequent analysis sections.

These sessions were held over several hours over both days of the summit and generally followed the same format: two formal presentations, followed by discussions (facilitated by a

**Table 1. Breakaway group session topics at national summit.**

| Title of breakaway group at national summit | Abbreviated title used in analysis |
|---|---|
| Group 1: Mental health promotion and prevention of mental disorders | Prevention & promotion |
| Group 2: Mental health research and innovation, and surveillance | Research & surveillance |
| Group 3: Mental health systems | Mental health systems |
| Group 4: Mental health infrastructure and human resources | Human resources & infrastructure |
| Group 5: Mental health and other conditions | Mental health & other conditions |
| Group 6: Mental Health Care Act of 2002 –lessons learned from implementation | Mental Health Care Act |
| Group 7: Child and adolescent mental health | Child & adolescent mental health |
| Group 8: Culture, faith-based practices and indigenous mental health practices | Culture & mental health |
| Group 9: Suicide prevention | Suicide prevention |
| Group 10: Advocacy, social mobilisation, user and community participation | Advocacy & user participation |

chair and captured by a rapporteur), during which groups formulated policy recommenda-tions. Group recommendations were fed back during a plenary session and in a closed-door meeting on Day 2 of the summit. Two documents were available for discussion during the group sessions: the draft mental health policy and a draft 'summit declaration' that, together with recommendations put forward at the summit, would form the official output of the national summit: The Ekurhuleni Summit Declaration (hereafter referred to as summit decla-ration). This was subsequently used in the finalisation of the policy document [20].

A previous study, described in [21], focused on procedural aspects of the consultation sum-mits, as well as detail regarding summit participants. Through analysis of key informant inter-views, policy documents, and consultation summit transcripts, it addressed the question of how the consultation process enabled or constrained movement of knowledge from enacted to inscribed forms. The findings of this study are detailed in a previous paper [21]. The analysis showed that no substantive changes were made to the mental health policy document follow-ing the consultation summits and suggested that the consultation may have been a rubber-stamping exercise, although this was not explicitly communicated to consultation participants [21].

The lack of substantive changes to policy may also have in part been a result of the lack of systematic processes for facilitating and capturing knowledge inputs from summit partici-pants, or for transferring these inputs between provincial and national levels [21]. The format of the consultation process limited participant interaction and the possibility for engagement with, or uptake of, more experiential forms of knowledge. The latter may have also in part been because of limited service user representation and facilitation of service user inputs [21]. Several procedural elements were found to limit the elicitation and transference of consulta-tion contributions for uptake into policy [21]. Some of these procedural elements will be drawn on in sections below to contextualise the subsequent findings. The current paper builds on this by focusing on the micro-level enactment and inscription of knowledge during the small group sessions at the national summit.

## Methods

### Study design, sampling and data collection

This descriptive, exploratory case study addressed the question of how participants' embodied knowledges were enacted and captured (inscribed) during a policy consultation process. Early in 2013, a request was sent to the South African national Department of Health (DoH) for the

records from the April 2012 national mental health consultation summit. The DoH sent audio-recordings of the 2-day summit, including recordings of the ten breakaway group sessions, which were subsequently transcribed verbatim. The focus of the current paper is on how embodied knowledge was enacted and inscribed during the breakaway group discussions at this summit; therefore, only the transcripts from these sessions, as well as the plenary session in which group recommendations were presented, are included here. A more detailed description of the summit context, procedural process, and participants has been provided in a previous paper [21].

## Data analysis

Data was analysed using thematic analysis [22] in NVivo. The principal author, a mental health professional who had experience in qualitative analysis of complex data sets in multi-country projects, conducted the analysis. Regular meetings were held with co-authors to discuss and verify both the analysis process and the application of themes (e.g., knowledge form codes) across the data set. The breakaway group transcripts were coded deductively, based on four lines of analysis: i) types of knowledge claims; ii) functions of knowledge claims; iii) responses to knowledge claims; and iv) inscription of knowledge claims. Each of these were coded according to pre-defined sets of coding rules, as follows:

i. **Types of knowledge claims** made during group discussions: experiential or evidence-based, both of which, according to Freeman and Sturdy's conceptual schema [7], are considered a form of embodied knowledge. Transcripts were first coded into two types of categories–presentations and discussions–according to when during the group session the talk occurred. The talk was then coded for type of knowledge–either as experiential, evidence-based, or other. (See coding framework in S1 Table).

ii. What participants were drawing on particular knowledge claims to *do*–in other words, **how knowledge claims were being used or enacted** during group discussions. The experiential and evidence-based knowledge claim types were further inductively coded for what participants were using these knowledge contributions to do within the discussion. The main themes arising in this analysis were: illustrate current situation (challenge/solution); highlight implications of a proposal (benefits/disadvantages); engage (support/counter). (See coding framework in S2 Table).

iii. Whether certain knowledge claims were more likely to be **responded to** during group discussions than others. Focusing on the discussion and the experiential and evidence-based categories above, the talk was then coded deductively for responsiveness, in three categories: responded to, not responded to, and response inaudible. Within the 'responded to' code, extracts were further according to whether they were 'responded to but not engaged with', versus 'responded to and further engaged with'. (See coding framework in S3 Table).

iv. Whether certain types of knowledge claims were more or less likely to be **'followed through' into inscribed form**–i.e. captured in the group recommendations. The experiential and evidence-based knowledge claim types were compared with the group recommendations presented by that group and coded as captured, partially captured, or not captured, depending on the extent to which the claims seemed to be aligned with a particular recommendation. (See coding framework in S4 Table).

Given the comparative focus of the analysis across multiple data sets, thematic content analysis was used as "a technique for categorising data and [estimating] the frequencies of these categories" [23] (p. 49). The analytic process followed for each of these four components was

broadly similar: group transcripts were coded either deductively or inductively according to the question being considered. They were then developed into composite codes that combined two or more coding categories (e.g. 'knowledge claim' + 'responded to'). These composite codes were analysed to determine frequencies (percentages) of particular types of knowledge claims within each area of interest and to allow for comparison across knowledge types, for example, the percentage of experiential knowledge claims that were responded to, **compared to** the percentage of evidence-based knowledge claims that were responded to.

### Ethical considerations

This study was approved in January 2013 by the Biomedical Research Ethics Committee (BREC) at the University of KwaZulu-Natal (ref: BE276/12), with annual renewals. It was considered to be of relatively low risk. All reasonable steps were taken to protect the confidentiality of participant responses through anonymisation of the data and aggregation of responses. Given the retrospective nature of the study, it was not possible to obtain individual consent for the use of the breakaway group transcripts. However, all participants at the summit were aware that the sessions were being audio-recorded and would be subsequently available through the Department of Health.

## Results

### Embodied knowledge claims

A total of 130 evidence-based knowledge claims and 95 experiential knowledge claims were identified. However, most of the talk (both presentations and discussions) could not be coded as either of these knowledge types; this content was thus coded as *Other*. Talk within this *Other* category varied considerably and including introduction and greetings, attempts by chairs at refocusing discussions, suggestions for changes to words or phrases in the draft documents, and so on. Examples of each of these types of evidence-based, experiential, and *Other* talk are included in S5 Table.

In order to contextualise the knowledge claims made during each group discussion, it is useful to note whether and how the formal presentations seemed to frame or influence the discussions. Further analysis was thus conducted to distinguish between when experiential and evidence-based knowledge claims were made during the formal presentations versus during the group discussions. The percentages of talk drawing on experiential, evidence-based, and *Other* knowledge claims in the breakaway group presentations and discussions are shown in Fig 1. The talk that did not draw explicitly on either evidence-based or experiential knowledge claims (i.e. *Other*) comprised the bulk (62%) of group presentations and discussions across all ten groups. Evidence-based knowledge claims (22%) were made slightly more frequently than experiential knowledge claims (16%).

### Enactment of knowledge claims

**Functions of knowledge claims: What were they being used to do?** A secondary level of analysis explored whether there appeared to be an association between particular types of knowledge claims (i.e., experiential and evidence-based) and the function they seemed to serve. The analysis here focused only on experiential and evidence-based knowledge claims made during the group discussions, as the formal presentations could be argued to be governed by pre-existing conventions regarding structure and content. There were thus 59 experiential knowledge claims and 32 evidence-based knowledge claims available for this analysis.

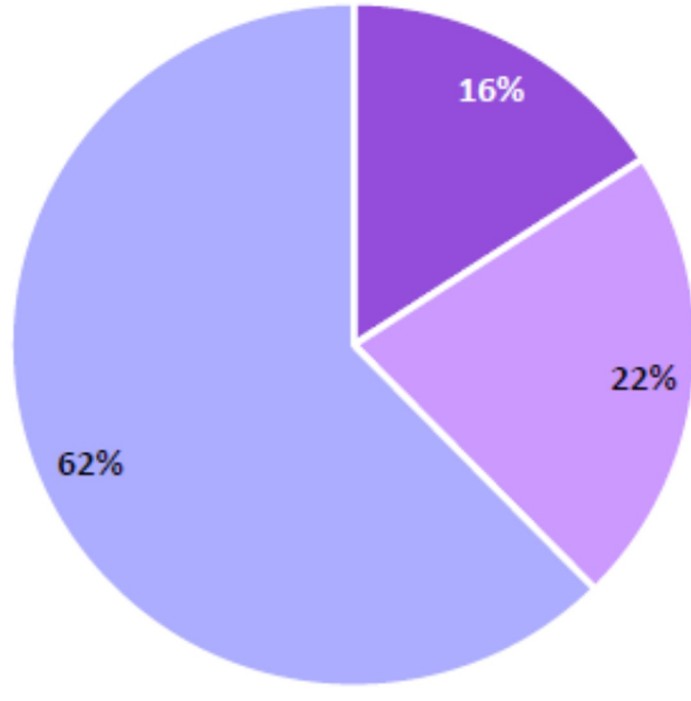

**Fig 1. Percentage of types of knowledge claims in breakaway groups.**

Three main functions for which both experiential and evidence-based knowledge claims seemed to be employed were identified. These were to illustrate a current ('on the ground') situation, to highlight the implications of or motivate for a particular proposal, and to engage with previous points made (see examples in S6 Table). The enacted uses of knowledge claims in the breakaway group sessions are shown in Fig 2.

Fig 2 shows that both experiential and evidence-based knowledge claims were explicitly drawn on to illustrate the current situation in relation to a particular issue (far left of Fig 2), more frequently than to perform any of the other functions identified. Of the instances identified where knowledge claims were used to illustrate a current situation, the majority of these were experiential knowledge claims, either illustrating a challenge or a solution/best practice. Similarly, where knowledge claims were used to highlight the implications of a particular proposal (centre of Fig 2), the majority of these were also experiential knowledge claims, either to highlight the benefits of or motivate for a proposal, or to highlight the disadvantages or argue against a proposal.

Conversely, evidence-based knowledge claims seemed to be used more frequently than experiential knowledge claims to engage with previous points made, either to support or to counter these points (far right of Fig 2). This lends partial support to the notion that evidence-based knowledge might be seen as a more valid and robust form of knowledge than experiential knowledge. Thus, in trying to substantiate arguments when engaging in debate with other group members, participants may be more likely to back up their arguments with evidence-based claims than with experiential claims. However, the difference between use of evidence-

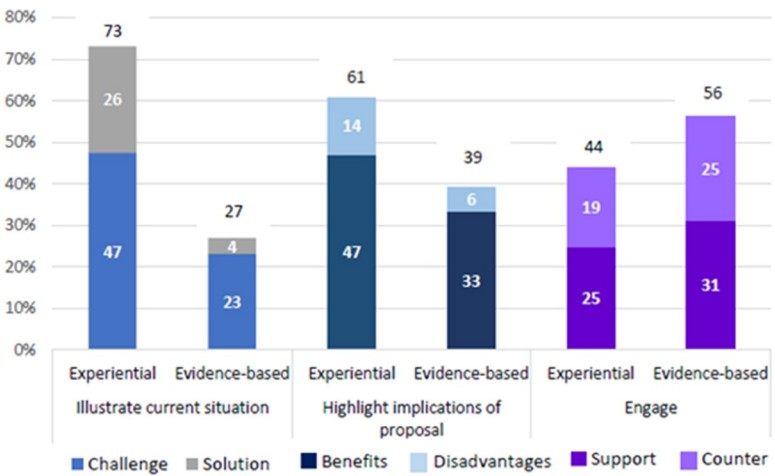

**Fig 2. Enacted uses of knowledge claims in breakaway group sessions.**

based knowledge claims and of experiential knowledge claims when engaging with others was too small to make stronger conclusions in this regard.

**Responses to knowledge claims: Were they responded to?** The enactment of knowledge in consultation spaces draws attention to the interactive nature of knowledge creation. In addition to the functions that evidence-based and experiential knowledge appeared to serve during these group discussions, then, the extent to which such contributions were responded to was also of interest. The references to experiential knowledge and evidence-based knowledge during the discussions were therefore coded according to whether these received a direct response. Where they were responded to, this was further differentiated between cases where the contribution was acknowledged, but not engaged with, and cases where the contribution was taken up and there was engagement. (See examples in S7 Table). Fig 3 illustrates the breakdown of responsiveness to evidence-based and experiential knowledge claims during breakaway group discussions.

As shown in Fig 3, both experiential and evidence-based knowledge claims were more frequently responded to than not. There did not seem to be a substantial difference, however, between experiential and evidence-based knowledge claims: both seemed equally likely to elicit some form of response or acknowledgement. This pattern is similar when looking at the extent to which different knowledge claims seemed to result in engagement. Where both types of knowledge claims elicited direct responses, the response was more frequently to engage with the points being made than merely to acknowledge an input (and move onto other points or participants). While those knowledge claims that were responded to were more likely to be engaged with more substantively than merely acknowledged, neither experiential nor evidence-based knowledge claims seemed more likely than the other to result in such engagement.

## Inscription of knowledge claims

Knowledge claims were coded as reflected or inscribed in group recommendations if the group recommendation was considered to closely reflect the underlying proposal contained in the particular knowledge claim. Partially reflected knowledge claims were coded as such where the group recommendation made reference to the general idea contained in the knowledge claim, but did not directly recommend what the speaker seemed to be proposing within that claim. Where there was no direct link to the knowledge claim in any of the recommendations put forward by the group, these knowledge claims were coded as not reflected.

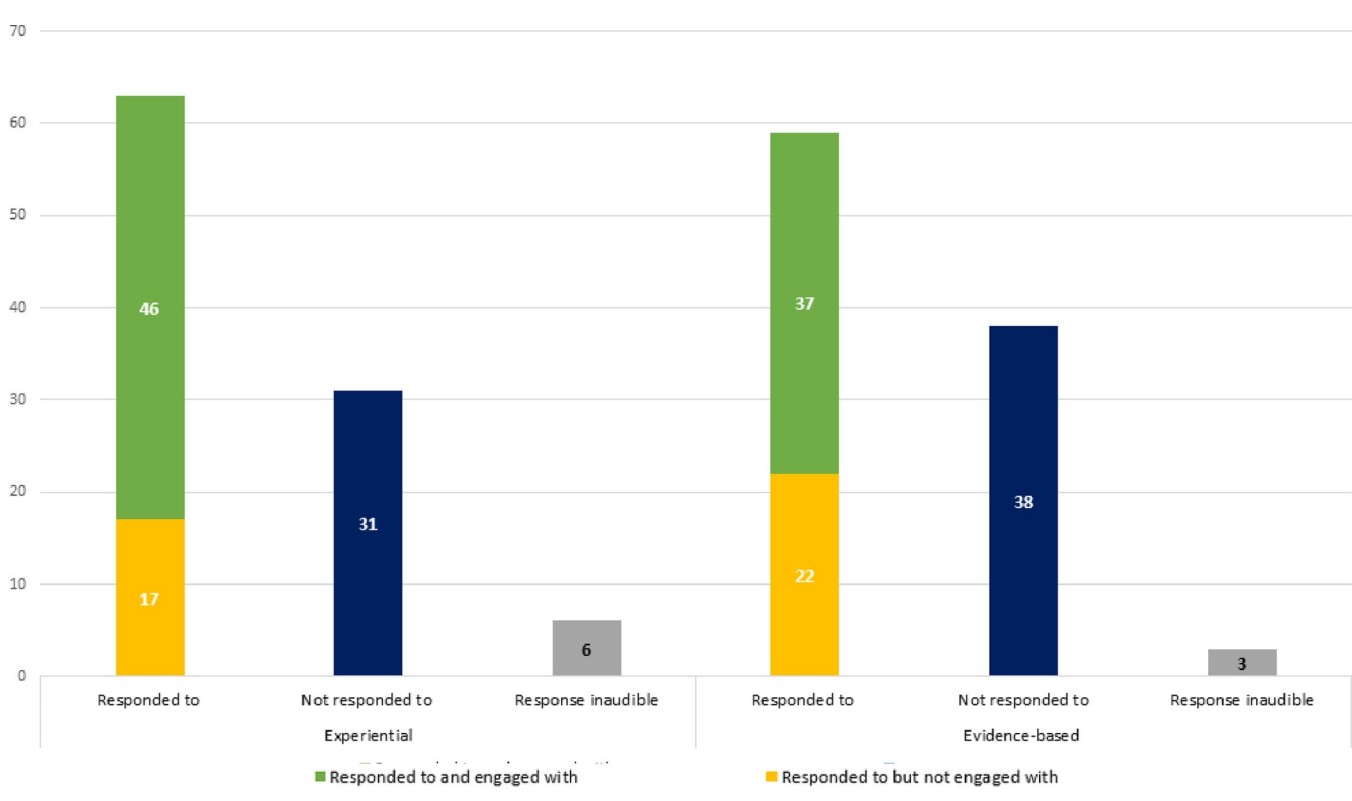

**Fig 3. Responsiveness to experiential and evidence-based knowledge claims.**

The percentage of experiential and evidence-based knowledge claims that were reflected, partially reflected, and not reflected in group recommendations is illustrated in Fig 4. There are roughly equal proportions of reflected, partially reflected, and not reflected claims across the two knowledge types, and in both cases, these knowledge claims were predominantly not reflected in group recommendations (66% and 61%, respectively). This suggests, firstly, that linking policy proposals to explicit knowledge claims did not seem to be associated with these inputs being take up into recommendations (although, as will be explored later, there are several possible reasons for this). Secondly, neither experiential knowledge claims nor evidence-based knowledge claims seemed more likely to be reflected in group recommendations than the other.

In summary, there was much less explicit reference to any type of knowledge claim to back up statements or proposals than might be expected during policy consultation discussions. The majority of talk during the ten breakaway group sessions at the national mental health summit did not draw explicitly on either evidence-based knowledge or experiential knowledge. Evidence-based knowledge claims were more often used to engage with previous points–whether to support or counter these–while experiential knowledge claims were more frequently used to illustrate current situations or highlight implications of proposals. Notably, there were no evident differences between the frequency with which either experiential or evidence-based knowledge claims were responded to or engaged with during group discussions. Neither evidence-based nor experiential knowledge claims appeared to be more likely to be reflected in group recommendations across all groups.

## Discussion

The micro level of policy consultation–where individuals engage with policy documents and in discussion with one another–provides a window on how embodied knowledge comes to be

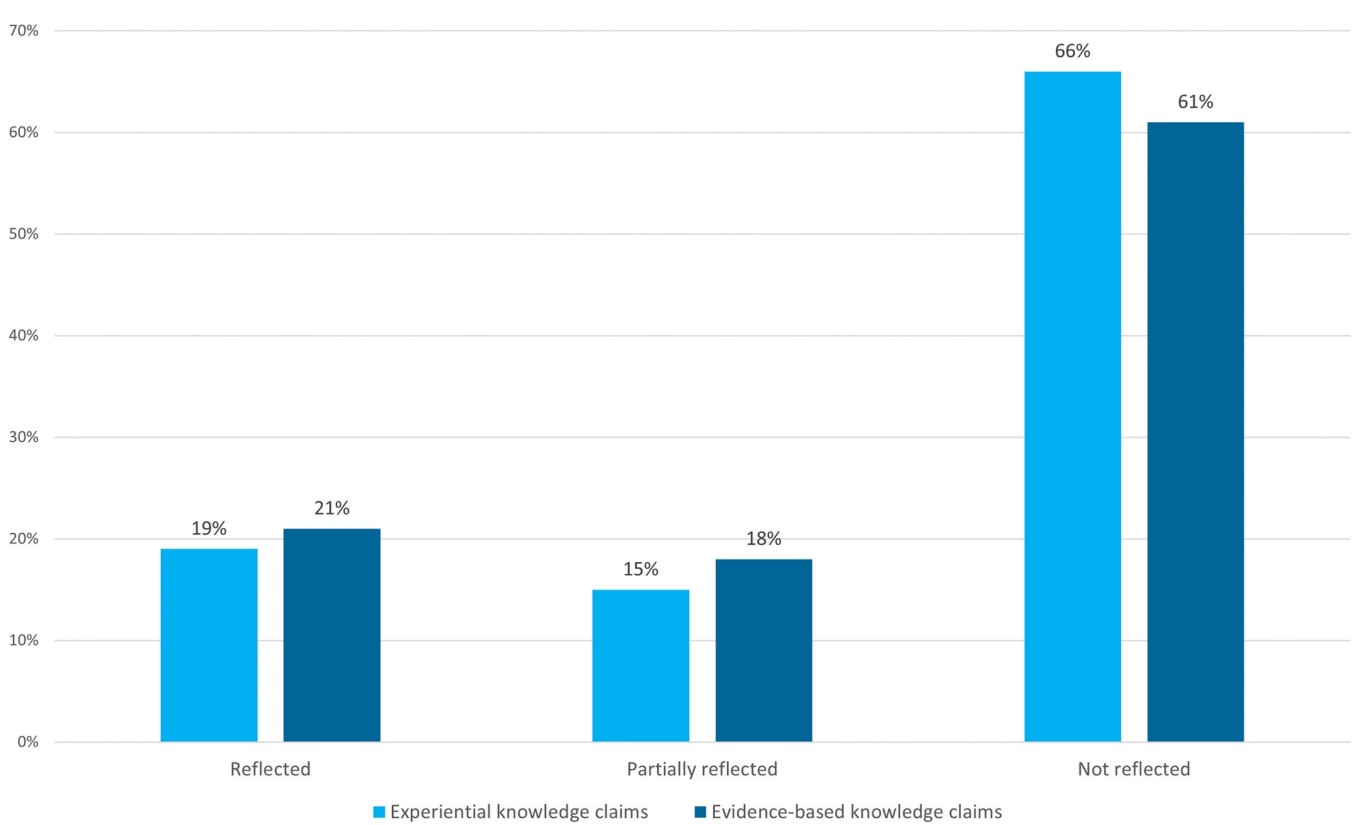

**Fig 4. Percentage of knowledge claims reflected in group recommendations.**

enacted in these spaces. In this study, neither evidence-based knowledge nor experiential knowledge was found to have been explicitly drawn on to substantiate claims for or against policy proposals. Neither of these forms of embodied knowledge appeared more likely than the other to have been responded to or inscribed in group recommendations.

Although these findings in some ways seemed to confirm the difficulties inherent in eliciting and capturing tacit embodied knowledge [19, 24], there were a number of surprising aspects in what was found. One was the relative lack of explicit reference to both evidence-based and experiential knowledge in these micro-level discussions. Another was that evidence-based knowledge did not seem to be favoured over experiential knowledge as a more legitimate knowledge source, which would have increased the likelihood of it being engaged with or recorded in recommendations. In the subsections that follow, several possibilities are explored to understand why this might have occurred.

## The exclusion of embodied experience

In contrast to studies showing that inputs from the public during policy consultation are more likely to be in embodied, experiential forms than in the factual or formal language of policy [15, 16], this study did not find this to be the case. This may be, in part, because those who came with more experiential or practical kinds of knowledge felt that they needed to align their inputs with more formal or explicit knowledge forms in order for them–and their contributions–to be seen as legitimate. This aligns with previous findings highlighting implicit demands on service-user participants to professionalise their talk [25, 26], or to translate their experiences into "actionable, bureaucratic vocabulary" [27] (p. 371) in order to be heard. The

efforts by Chairs to reframe participant inputs in the form of indicator-linked proposals or recommendations [28] may also have contributed to this. This speaks to the side-lining of service-user or lay participants through the silencing of embodied knowledge during policy consultation processes, echoing what Smith-Merry found to be "new ambivalences that were emerging around the role of embodied knowledge within the mental health policy sphere" [29] (p. 36).

The poor representation of mental health care service users at the national consultation summit [28] may be another reason for the observed lack of explicit experiential knowledge drawn on during group discussions. This confirms other studies demonstrating a lack of service-user participation in policy development in LMICs [13, 30], and in South Africa in particular [12, 31]. It was also evident in this study [28], as in other studies [29, 32], that the limited service-user involvement in the consultation summit appeared to be somewhat tokenistic. This situation risks perpetuating the negative perceptions and stigma regarding service users from which such tokenistic involvement stems [12].

Thus, one possibility for the infrequency of explicit claims in reference to embodied knowledge found in this study is the potential (and ongoing) side-lining and silencing of mental health care service users' inputs. There are implications in this regarding the implicit 'rules of the game' according to which participants are expected to contribute in specific ways to policy consultation processes. However, the finding that there was equally infrequent reference to evidence-based knowledge during the consultation discussions suggests that it was not just experiential embodied knowledge that was disallowed, whether implicitly or explicitly. What got responded to and inscribed in enacted spaces in this case study seemed somewhat arbitrary, given that much of the talk in group discussions was classified as *Other* (for example, reiterating or countering statements made by others, or generic observations or opinions), with no explicit reference to knowledge claims to substantiate statements or justify proposals. Looking more closely at how knowledge was enacted might shed some light on why this was the case.

## The implicit rules of enactment

The implicit rules of the game [33] connected to the specific format adopted at this consultation summit may have allowed or disallowed particular forms of knowledge to be enacted. In addition, the "language games" [34] (p. 181) inherent in such participatory processes allow for particular ways in which embodied knowledge can be enacted. This aligns with previous work showing that different forms of participation affect the ways in which people are able to participate [35, 36].

The framing of each breakaway group session with formal presentations by experts in the relevant field may have already laid the 'evidence' groundwork around which contributions could take place. This certainly seemed to be the case for those groups in which participants used both the presentation and presenter as reference points throughout the discussion. Thus, participants allowed these presentations to do the work of evidence-based knowledge, such as legitimating certain proposals. It may also have been the case that these expert presentations served to communicate certain rules according to which contributions could be made, perhaps even *dis*allowing evidence-based claims from individual participants. In this way, the implicit rules of enactment may have served as a form of surveillance and sanctioning that monitored and regulated how knowledge could be enacted in these formal conventionalised settings [7].

## The assumption of shared knowledge

Appeals to both evidence and experience have been shown to be useful in persuading others on the merits of an argument or legitimacy of a position [37–40]. The relative lack of such

claims in this study stands in contrast to a number of other studies which have identified the use of explicit claims to motivate for particular positions or proposals. Cheng and Johnstone, for example, found that people appealed to their own personal experience and gave reasons for policy claims much more often than for other forms of claims [41]. In her study on knowledge and knowing in policy work, Maybin found that policymakers make pragmatic use of knowledge claims–facts, figures and stories–to "generate support for policies and to defend decisions taken" [42] (p. 3).

However, Maybin's study was focused on policymakers themselves, and on how they used knowledge in their everyday work [42]. The current study's finding that neither evidence-based nor experiential knowledge claims were frequently referenced during consultation discussions could also be because it was not necessary for participants to make these knowledge claims explicit. One possible explanation might be the implicit assumption of shared knowledge. Antaki and Leuder studied the role of "claim-backing–the use of explanations to warrant the truth of what is said" in rhetoric and argumentation–and suggested that "the way people back claims reveals the mutual knowledge on which their appeals succeed or fail" [43] (p. 279). While the point of making explicit claims or giving reasons might be to justify a particular position, Adams suggests that this may not be necessary if speakers and listeners share the same background [37]. He argues that "when talking with those who are similar to us, we can get away with implying large parts of our argument because others will be able to fill in the blanks" [37] (p. 2).

There were indications that, while there were few service users at the summit, there were greater numbers of academics/researchers and practitioners–or individuals who occupied both roles [28]. This was evident in the introductions that some groups engaged in during group sessions; it was also seen in the programme, as well as this study's interview participants' observations that particular voices or perspectives were strongly represented in both the policy and the implementation plan [28]. It might be fair to assume, then, that the groups comprised somewhat familiar communities of practice–or, in Freeman and Sturdy's terms, "communities of knowers"–and that participants did not feel it necessary to link their statements or proposals back to explicit knowledge claims [7] (p. 14).

## An emphasis not on evidence-based knowledge over experiential knowledge but on enacted and inscribed knowledge forms

While evidence-based and experiential knowledge claims were not clearly evident in these findings, what emerged more clearly was that emphasis was placed on some *phases* of knowledge more than on others. Specifically, the emphasis was on the somewhat rapid transformation of enacted knowledge into inscribed knowledge. In other words, the pressure to produce recommendations and the limited time available to do this [28] meant that the knowledge that was enacted during group discussions tended to be quickly transformed into a form that made it more amenable to inscription. This, together with what has been discussed above, might explain the relative lack of explicitly referenced experiential and evidence-based knowledge. It also might explain the surprising finding that the traditional evidence-experience dichotomy was not present.

Instead, it suggests that attending to phases of knowledge may be more important than being concerned with the legitimacy of evidence-based or experiential knowledges. In particular, if the goal of consultation is to draw on the knowledge that participants bring, more time and space need to be provided for enacted knowledge forms, before attending to how this enacted knowledge might be captured in inscribed form. It nonetheless still points to–perhaps even more so–the importance of creating processes that enable not only the enactment of embodied

knowledge, but also the interactions and transformations between the three knowledge forms: embodied, enacted, and inscribed knowledge. In order to give input on draft policies, consultation participants must be afforded the chance to properly engage with the inscribed knowledge of this policy during discussions; this, in turn, might produce enacted knowledge that is of greater value to policymakers than embodied or inscribed knowledge alone.

To some extent, the focus on inscribed knowledge speaks to the purpose of policy consultation–the necessity of gathering and capturing a large amount of input. However, it seems that, in this particular consultation process at least, more attention was paid to the *what* of what was captured than to the *how* of capturing it, and the extent to which the latter might have optimised use of valuable knowledges. In similar ways, embodied knowledge did not seem to move easily through the mental health consultation process in Smith-Merry's research which, she suggests, may have been because "crucial stages of the consultation entailed or relied on inscription" [29] (p. 35).

This highlights the importance of attending to the forms of knowledge enactment and inscription and, more specifically, to the points at which one form of knowledge is transformed into another. In the absence of this, both the conventional orientation towards inscription in policy consultation processes [19, 29], as well as towards particular kinds of inscribed knowledge forms (e.g. formal reports), may marginalise "some types of knowledge from policy processes and, hand in hand with this, the identities and needs of those who hold that knowledge" [29] (p. 38) Attending to the enactment and inscription of experiential embodied knowledge is particularly important in the context of mental health [16].

## Limitations

Several limitations may have influenced the results of this study. Although in most cases there were audible audio-recordings for the ten breakaway group sessions, there were instances in which a particular session's recording was either not available or the audibility of a particular recording was poor. Conclusions regarding the nature of knowledge inputs should thus be interpreted with caution. The extent to which knowledge claim contributions were responded to or not seemed to be quite strongly dependent on the person who was chairing or facilitating the discussions. In most cases where there were responses, these were primarily made by the group chair or facilitator; there was also limited time in which to follow through in depth about any particular point or proposal. It is therefore possible that the degree to which particular knowledge claims were responded to in such cases may not have been (only) related to the form in which the contribution was made.

It is not possible to draw direct or causal links where particular types of knowledge claims were found to be inscribed in group recommendations, as this may be attributed to many other factors. Where particular knowledge claims are clearly shown *not* to be reflected in group recommendations, stronger conclusions might be drawn, although, again, the extent to which this can be attributed to the framing of the knowledge claims as opposed to other influencing factors is limited. It is neither feasible nor even desirable to capture all knowledge contributions into consultation outputs. However, the premise of this study has been that if policy consultation is undertaken, this should be done with a genuine intention to attend to and consider knowledge inputs during these processes, particularly to avoid it being a tokenistic exercise.

## Conclusion and recommendations

Attending to the specific ways in which knowledge is transformed and moved through a policy consultation process has the potential to enhance the value that consultation offers policymakers in policy formulation. This study used Freeman and Sturdy's *embodied-enacted-*

*inscribed* knowledge framework [7] to explore how a particular mental health policy consultation process informed policy by tracing the movement of different forms of knowledge through the process. A key insight from this study is that it is not only different types of knowledge–such as evidence-based or experiential–that must be reconciled in policy(making), but also different forms (embodied, enacted, and inscribed). Just as evidence-based and experiential types of knowledge have particular advantages and disadvantages with respect to their role in policy, so too do embodied, enacted, and inscribed knowledges. Placing emphasis on one form–such as inscribed knowledge–risks limiting the potential value that other forms have to offer policy and policymakers. A recommendation for consultation participants is to make contributions that are explicitly linked to both the content of draft policy documents as well as to knowledge claims that would allow for such contributions to be warranted. In addition, this study highlights the importance of attending to the points at which knowledge transforms between embodied, enacted, and inscribed forms. This, in turn, highlights the importance of designing participatory processes that enable optimal use of knowledge inputs in these enacted spaces, in order to align assumptions about the value of policy consultation with consultation practices. It is recommended that instructions to consultation participants be much clearer and should involve greater inputs by participants themselves as to how best to navigate through the intersections of *embodied-enacted-inscribed* knowledge. There should be upfront and explicit agreement on making knowledge contributions, as well as how these will be managed and captured during the process. Organisers of consultation processes should consider different forms of participatory designs that would enable the enactment and inscription of a variety of knowledge inputs. One form of participatory process that holds promise is the open-space design format. This can accommodate large numbers of people while simultaneously allowing for greater interaction between participants, as well as engagement with policy proposals on the table.

This study has shown that the *embodied-enacted-inscribed* knowledge framework [7] has been a useful lens through which to make sense of the use and movement of knowledge through a policy consultation process in a LMIC context. Future research may include qualitative studies that explore participants' and policymakers' perspectives on the utility of thinking of contributions to policy consultation in this way in order to optimise contributions. The value that different types of knowledge (e.g. evidence-based and experiential knowledge) might offer at different points in the policymaking process has been emphasised in this paper. It is important, however, to be able to assess the credibility of more experiential and practical knowledge forms, while optimising the value they add. Research is needed to develop and test possible credibility criteria against which knowledge inputs that have not been empirically tested might be evaluated.

## Supporting information

**S1 Table. Types of knowledge claims coding framework.**
(DOCX)

**S2 Table. What knowledge claims were being used to do coding framework.**
(DOCX)

**S3 Table. Responses to knowledge claims coding framework.**
(DOCX)

**S4 Table. Reflection / inscription of knowledge claims in group recommendations coding framework.**
(DOCX)

**S5 Table. Examples of knowledge claim types.**
(DOCX)

**S6 Table. Examples of functions of knowledge claims.**
(DOCX)

**S7 Table. Examples of responses to knowledge claims.**
(DOCX)

## Author Contributions

**Conceptualization:** Debra Leigh Marais, Michael Quayle.

**Data curation:** Debra Leigh Marais.

**Formal analysis:** Debra Leigh Marais.

**Methodology:** Debra Leigh Marais.

**Supervision:** Inge Petersen, Michael Quayle.

**Writing – original draft:** Debra Leigh Marais.

**Writing – review & editing:** Inge Petersen, Michael Quayle.

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
