## [Decision Letter · Decision Letter 0]

9 Dec 2020

PONE-D-20-22171

Policymaking through a knowledge lens: Using the embodied-enacted-inscribed knowledge framework to illuminate the transfer of knowledge in a mental health policy consultation process – a South African case study

PLOS ONE

Dear Dr. Marais,

Thank you for submitting your manuscript to PLOS ONE. After careful consideration, we feel that it has merit but does not fully meet PLOS ONE’s publication criteria as it currently stands. Therefore, we invite you to submit a revised version of the manuscript that addresses the points raised during the review process.

It has been a challenge finding reviewers - apologies for the delay. However, The academic editor agrees with the need for conclusion that minor revisions are needed if the manuscript is to be considered for publication.. 

We look forward to receiving your revised manuscript.

Kind regards,

Joseph Telfair, DrPH, MSW, MPH

Academic Editor

PLOS ONE

Journal Requirements:

Reviewers' comments:

Reviewer's Responses to Questions

**Comments to the Author**

1. Is the manuscript technically sound, and do the data support the conclusions?

Reviewer #1: Yes

2. Has the statistical analysis been performed appropriately and rigorously? 

Reviewer #1: N/A

3. Have the authors made all data underlying the findings in their manuscript fully available?

Reviewer #1: No

4. Is the manuscript presented in an intelligible fashion and written in standard English?

Reviewer #1: Yes

5. Review Comments to the Author

Reviewer #1: This was an interesting and well-written paper that examines the increasing democratization of policymaking as well as the growing importance of service user involvement in mental health policy development and implementation. The authors’ in-depth examination of the processes of knowledge transformation during a policy consultation process has implications not only for policymaking but also for other instances of service user or public involvement (e.g. in service organization and even research). Overall the methodology was sound and the literature was sufficiently taken into consideration. My comments are intended to further strengthen the paper and clarify certain methodological details.

Introduction:

- The introduction was well written but one element that could be improved is to have a single and clear aims statement. Currently, the authors state their aims at the bottom of page 5 (lines 116-118) but also at the beginning of the section “Context of the study” (lines 163-164). I propose that the authors reformulate their sentence at the beginning of the context section to ensure there is no confusion as to the main aims of the study.

Methods:

- The authors provide adequate description of their data sources and analysis process but there is very little information provided about the participants of the summit and each of the breakout sessions. We learn only in the discussion that there appeared to be few service users in attendance but the composition of participants in the breakout sessions should be reported, as it will influence the types of knowledge claims made during these sessions. For instance, were there service users present in every breakout session or only some sessions? How many?

- Similarly, in the description of the national summit and breakout sessions some details could be provided in the context section on the chairs/facilitators of the breakout sessions and the context within which the discussions took place (e.g. time constraints).

- One element missing from the description of the analysis is which research team members participated in the analysis and how this was done. Was the analysis carried out by a single person or a team, with what experience or background, etc.

- Small detail, but given how the results are reported, it would seem sensible when presenting the four lines of analysis (page 10, line 214) to re-order the information so that the functions of knowledge claims appears before the responses to knowledge claims.

Results/Discussion:

- One of the authors’ main findings is that participants’ knowledge claims (whether evidence-based or experiential) were largely not reflected in the group recommendations made during the summit’s plenary session. The authors suggest that this indicates that participants’ knowledge claims may not be sufficiently captured and inscribed during the consultation process, which in theory runs contrary to the purpose of the consultation process itself. However, one thing that was less clear was whether the national summit was held principally as a way to validate and ensure acceptability of policy directions determined by the government or whether the goal really was further development of these policies. If the goal was the former, it would explain why only a small number of knowledge claims were considered during the plenary discussions, as the goal may have been to engage in a consultation exercise without actually challenging the policies too much. In other words, was the government’s approach more political than instrumental? Were the goals of the national summit explicitly stated in any documentation surrounding the event?

- Another related issue is that since the paper’s focus is on the micro events during the national summit, we have little information on how the earlier provincial summits informed the draft policy that was presented during the national summit or to what extent the final policy plan (post summits) may have been informed by different forms of knowledge during the development process. It would be reasonable to think that there may have been a more complete capture and transformation of knowledge during the initial rounds of consultations (when the policies really were being developed) than during the national summit when the government had already determined its policy directions and the justification for them. The paper would have us conclude that the government may not have used appropriate methods to capture the views of different stakeholders, but can that conclusion really be made without considering the information and knowledge shared by participants during those provincial summits?

- On page 18 the authors state that it is evident that the service user involvement in the national summit appeared tokenistic, but they provide no real justification for this statement aside from the limited number of service user participants. What makes the authors think this? Were the service user knowledge claims treated differently than those of other participants?

6. PLOS authors have the option to publish the peer review history of their article (what does this mean?). If published, this will include your full peer review and any attached files.

Reviewer #1: **Yes: **Matthew Menear

---

## [Author Response · Author response to Decision Letter 0]

12 Dec 2020

Academic Editor's feedback

The style guide and file naming requirements have been carefully attended to and the manuscript and corresponding files amended accordingly. 

2. We note that you have indicated that data from this study are available upon request. 

Data cannot be publicly shared because there are ethical restrictions on sharing the complete data set. The data comprises verbatim transcripts of discussions in each of the ten breakaway groups, with a particular focus on Chairs’ input. Although permission was obtained by the Department of Health to record these sessions, this was not explicitly provided to use the information for research purpose. The group of participants participating in these discussions, and the Chairs of these sessions, would be potentially identifiable from their narratives and the way they spoke about their experience and positions as part of their input to the policy discussions, despite the use of participant codes to de-identify the data. Sharing the full data set would violate the principle of confidentiality.

For researchers who meet the criteria for access to confidential data, the Biomedical Research Ethics Committee at the University of KwaZulu-Natal (South Africa) can be contacted to request permission to access the data (contact: Ms Anusha Marimuthu, +27312604769, BREC@ukzn.ac.za).

3. Please include captions for your Supporting Information files at the end of your manuscript, and update any in-text citations to match accordingly. 

Captions for Supporting Information files have been added at the end of the manuscript. In-text labels and file names have been updated according to the guidelines. 

Comments from Reviewer 1

1. Introduction:

The introduction was well written but one element that could be improved is to have a single and clear aims statement. Currently, the authors state their aims at the bottom of page 5 (lines 116-118) but also at the beginning of the section “Context of the study” (lines 163-164). I propose that the authors reformulate their sentence at the beginning of the context section to ensure there is no confusion as to the main aims of the study.

The aim in the Context section has been rephrased to align with the earlier stated aim. 

2. Methods:

The authors provide adequate description of their data sources and analysis process but there is very little information provided about the participants of the summit and each of the breakout sessions. We learn only in the discussion that there appeared to be few service users in attendance but the composition of participants in the breakout sessions should be reported, as it will influence the types of knowledge claims made during these sessions. For instance, were there service users present in every breakout session or only some sessions? How many?

A full list of summit participants was not available (or perhaps even recorded). A foreword of the final policy document states that the summits were attended by representatives from “research groups, academia, professional associations and statutory health institutions, the World Health Organization, nongovernmental organizations, mental health care user groups, clinicians, national and provincial departments that play a role in mental health.”. 

Similarly, a list of participants at each breakaway group was either not recorded or not made available by the Department of Health. Summit participants self-selected into different breakaway groups. It was possible to identify many of the speakers from the audio-recordings as they were either identified by the Chair or self-identified. Most of these were either academics or service providers (or both), although more service users were identified in Group 10 (on Advocacy, social mobilisation, user and community participation). 

These contextual issues (including participant details) have been described in an earlier paper. 

In addition, the lack of representation of service users at the summit was identified as an issue by representatives from service user groups who were interviewed during an earlier phase of this study. These findings are reported on in the earlier paper. 

A reference to a more detailed description of the context and participants in the summit being provided in an earlier paper has been included in the Methods section of this paper.

3. Methods: 

Similarly, in the description of the national summit and breakout sessions some details could be provided in the context section on the chairs/facilitators of the breakout sessions and the context within which the discussions took place (e.g. time constraints).

These procedural issues (including facilitation by Chairs) were reported on in the previous paper referenced above.

Reference to the description of procedural issues in the previous paper has been included in the Methods section of this paper. 

4. Methods:

One element missing from the description of the analysis is which research team members participated in the analysis and how this was done. Was the analysis carried out by a single person or a team, with what experience or background, etc.

The principal author, a mental health professional who had experience in qualitative analysis of complex data sets in multi-country projects, conducted the analysis. Regular meetings were held with co-authors to discuss and verify both the analysis process and the application of themes (e.g., knowledge form codes) across the data set. This detail has been included in the Methods (data analysis) section. 

5. Methods: 

Small detail, but given how the results are reported, it would seem sensible when presenting the four lines of analysis (page 10, line 214) to re-order the information so that the functions of knowledge claims appears before the responses to knowledge claims.

Thank you for this useful feedback. The “responses to” and “functions of” categories have been swapped around in the Methods section to correspond with the Results section. 

6. Results/Discussion:

One of the authors’ main findings is that participants’ knowledge claims (whether evidence-based or experiential) were largely not reflected in the group recommendations made during the summit’s plenary session. The authors suggest that this indicates that participants’ knowledge claims may not be sufficiently captured and inscribed during the consultation process, which in theory runs contrary to the purpose of the consultation process itself. However, one thing that was less clear was whether the national summit was held principally as a way to validate and ensure acceptability of policy directions determined by the government or whether the goal really was further development of these policies. If the goal was the former, it would explain why only a small number of knowledge claims were considered during the plenary discussions, as the goal may have been to engage in a consultation exercise without actually challenging the policies too much. In other words, was the government’s approach more political than instrumental? Were the goals of the national summit explicitly stated in any documentation surrounding the event?

It is stated in the forewords in the final policy itself that the purpose of the provincial and national summits was to develop and finalise the draft policy. A previous paper described interviews with participants at both the provincial and national summits, describing how the same draft policy was available at the national summit as had been at the provincial summit. This paper also reports that a draft of the ‘summit resolution’ from the national summit was already available to participants before discussions began. Some interviewees had been convenors of the provincial and national summit themselves and reported having no clear direction regarding how inputs should be captured or used, as well as a sense that the consultation process was a rubberstamping exercise – a purpose that was not made clear to the over 4000 participants in the provincial and national summits, some of whom were interviewees who reported feeling misled. If the purpose of the summits was to endorse the policy, there was still no systematic process for capturing participants’ inputs from across the breakaway group discissions in order to check this against the existing policy and confirm this, which was particularly important given that no government officials mandated with the policy were not in attendance at these discussions. These findings are reported on in this previous paper. 

A reference to the lack of clear purpose described in the previous paper has been added to the Context section of this paper. 

7. Results/Discussion: 

Another related issue is that since the paper’s focus is on the micro events during the national summit, we have little information on how the earlier provincial summits informed the draft policy that was presented during the national summit or to what extent the final policy plan (post summits) may have been informed by different forms of knowledge during the development process. It would be reasonable to think that there may have been a more complete capture and transformation of knowledge during the initial rounds of consultations (when the policies really were being developed) than during the national summit when the government had already determined its policy directions and the justification for them. The paper would have us conclude that the government may not have used appropriate methods to capture the views of different stakeholders, but can that conclusion really be made without considering the information and knowledge shared by participants during those provincial summits?

The similar absence of systematic processes at the provincial summits and a lack of follow through between provincial and national summit has been reported on in a previous paper. The draft policy that was available at the start of the national summit was the same draft policy that was available at the start of the provincial summits, suggesting that the inputs at the provincial summits had served the same purpose as those at the national summit. This is reported on in the previous paper. 

References to these process issues and issues of follow through between provincial and national summits described in this previous paper have been added to the Context section. 

8. Results/Discussion:

On page 18 the authors state that it is evident that the service user involvement in the national summit appeared tokenistic, but they provide no real justification for this statement aside from the limited number of service user participants. What makes the authors think this? Were the service user knowledge claims treated differently than those of other participants?

As mentioned earlier, it was possible to discern that the majority of speakers giving input in the discussions were academics and/or service providers. 

The lack of service user representation and the way in which service user contributions were treated in a ‘tokenistic’ manner have been reported on in a previous paper, where service user representatives who had been involved in organising and/or participating in the summits reported these issues. 

Reference to the previous paper in which this limited and tokenistic service user representation is described has been included in the Context section of this paper.

---

## [Decision Letter · Decision Letter 1]

21 Dec 2020

Policymaking through a knowledge lens: Using the embodied-enacted-inscribed knowledge framework to illuminate the transfer of knowledge in a mental health policy consultation process – a South African case study

PONE-D-20-22171R1

Dear Dr. Marais,

We’re pleased to inform you that your manuscript has been judged scientifically suitable for publication and will be formally accepted for publication once it meets all outstanding technical requirements.

Kind regards,

Joseph Telfair, DrPH, MSW, MPH

Academic Editor

PLOS ONE

Additional Editor Comments (optional):

Reviewers' comments:

Reviewer's Responses to Questions

**Comments to the Author**

1. If the authors have adequately addressed your comments raised in a previous round of review and you feel that this manuscript is now acceptable for publication, you may indicate that here to bypass the “Comments to the Author” section, enter your conflict of interest statement in the “Confidential to Editor” section, and submit your "Accept" recommendation.

Reviewer #1: All comments have been addressed

2. Is the manuscript technically sound, and do the data support the conclusions?

Reviewer #1: Yes

3. Has the statistical analysis been performed appropriately and rigorously? 

Reviewer #1: N/A

4. Have the authors made all data underlying the findings in their manuscript fully available?

Reviewer #1: No

5. Is the manuscript presented in an intelligible fashion and written in standard English?

Reviewer #1: Yes

6. Review Comments to the Author

Reviewer #1: (No Response)

7. PLOS authors have the option to publish the peer review history of their article (what does this mean?). If published, this will include your full peer review and any attached files.

Reviewer #1: **Yes: **Matthew Menear

---

## [Editor Report · Acceptance letter]

23 Dec 2020

PONE-D-20-22171R1 

Policymaking through a knowledge lens: Using the *embodied-enacted-inscribed* knowledge framework to illuminate the transfer of knowledge in a mental health policy consultation process – a South African case study 

Dear Dr. Marais:

I'm pleased to inform you that your manuscript has been deemed suitable for publication in PLOS ONE. Congratulations! Your manuscript is now with our production department. 

Kind regards, 

on behalf of

Dr. Joseph Telfair 

Academic Editor

PLOS ONE